# False Liver Metastasis by Positron Emission Tomography/Computed Tomography Scan after Chemoradiotherapy for Esophageal Cancer—Potential Overstaged Pitfalls of Treatment

**DOI:** 10.3390/cancers16050948

**Published:** 2024-02-26

**Authors:** Sen-Ei Shai, Yi-Ling Lai, Chen-I Chang, Chi-Wei Hsieh

**Affiliations:** 1Department of Thoracic Surgery, Taichung Veterans General Hospital, Taichung 40705, Taiwan; linglai@vghtc.gov.tw; 2Department of Applied Chemistry, National Chi Nan University, Nantou 545301, Taiwan; 3Institute of Clinical Medicine, National Yang-Ming Chiao-Tung University, Taipei 112304, Taiwan; 4School of Medicine, National Yang-Ming Chiao-Tung University, Taipei 112304, Taiwan; jenny28950.md07@nycu.edu.tw; 5School of Medicine, National Cheng Kung University, Tainan 701401, Taiwan; i54091149@gs.ncku.edu.tw

**Keywords:** false liver metastasis, neoadjuvant chemoradiotherapy (nCRT), F-18-fluorodeoxyglucose (18F-FDG), positron emission tomography–computed tomography (PET-CT), radiation-induced liver injury (RILI), radiation-induced liver disease (RILD)

## Abstract

**Simple Summary:**

FDG PET-CT scans are critical in detecting metastases during neoadjuvant chemoradiotherapy for esophageal cancer, particularly for potential liver involvement. The liver’s proximity to the radiation field in distal esophageal cancer therapies raises the risk of radiation-induced liver damage. Therefore, greater FDG absorption in the liver does not always imply metastases; it could also signal radiation-induced damage, which is a concern for distal esophageal carcinoma therapies in the left hepatic lobe, potentially leading to overstaging. Accordingly, thorough monitoring of FDG activity in the liver is required to reliably distinguish between radiation effects and genuine distant metastases. If FDG activity is seen in the left or caudate liver lobes following CRT, additional diagnostic procedures are demanded to confirm or rule out distant metastases. Surgery, usually scheduled 6–8 weeks after CRT, should be followed by an FDG PET-CT scan to look for new interval metastases, as their existence may prohibit surgical intervention.

**Abstract:**

In patients with esophageal cancer undergoing neoadjuvant chemoradiotherapy (nCRT), subsequent restaging with F-18-fluorodeoxyglucose (18F-FDG) positron emission tomography–computed tomography (PET-CT) can reveal the presence of interval metastases, such as liver metastases, in approximately 10% of cases. Nevertheless, it is not uncommon in clinical practice to observe focal FDG uptake in the liver that is not associated with liver metastases but rather with radiation-induced liver injury (RILI), which can result in the overstaging of the disease. Liver radiation damage is also a concern during distal esophageal cancer radiotherapy due to its proximity to the left liver lobe, typically included in the radiation field. Post-CRT, if FDG activity appears in the left or caudate liver lobes, a thorough investigation is needed to confirm or rule out distant metastases. The increased FDG uptake in liver lobes post-CRT often presents a diagnostic dilemma. Distinguishing between radiation-induced liver disease and metastasis is vital for appropriate patient management, necessitating a combination of imaging techniques and an understanding of the factors influencing the radiation response. Diagnosis involves identifying new foci of hepatic FDG avidity on PET/CT scans. Geographic regions of hypoattenuation on CT and well-demarcated regions with specific enhancement patterns on contrast-enhanced CT scans and MRI are characteristic of radiation-induced liver disease (RILD). Lack of mass effect on all three modalities (CT, MRI, PET) indicates RILD. Resolution of abnormalities on subsequent examinations also helps in diagnosing RILD. Moreover, it can also help to rule out occult metastases, thereby excluding those patients from further surgery who will not benefit from esophagectomy with curative intent.

## 1. Introduction

Esophageal cancer is responsible for over 450,000 deaths annually, ranking as the sixth leading cause of cancer-related mortality worldwide [1]. Surgical resection of the esophagus, following neoadjuvant chemoradiotherapy (nCRT), is the standard of treatment for patients with non-metastasized esophageal cancer [1,2,3]. nCRT has been shown to downstage tumors and increase the rate of radical resection, and it is associated with improved survival outcomes [2,3]. Esophageal cancer frequently metastasizes lymph nodes in the abdomen, liver, and lungs [4], with liver metastases in up to 35% of cases [5]. Several studies have evaluated the use of F-18-fluorodeoxyglucose (18F-FDG) positron emission tomography–computed tomography (PET-CT) for pre-surgical restaging after nCRT, finding varying rate of interval metastases ranging from 2% up to 26% [6,7]. However, these studies often lacked diagnostic accuracy measures, such as sensitivity and specificity, and included only a small patient cohort [8,9,10]. Accurate preoperative identification of (interval) metastasis is essential for selecting suitable candidates for surgery [11]. Radiation therapy for liver and upper abdominal perihepatic tumors has been limited due to the liver’s radiosensitivity [12]. Radiation damage, especially to the liver’s lateral segment near the distal esophagus, is challenging to avoid [13]. Preoperative whole-body FDG PET-CT is routinely used to assess the radiation response and exclude metastases. It can detect radiation-induced liver damage 2 to 6 weeks after therapy, as indicated by increased FDG uptake and reduced CT attenuation [13]. Radiation-induced liver disease (RILD), a significant complication, can present in classic or non-classic forms, characterized by diffuse or focal FDG uptake on PET-CT [14]. Understanding RILD’s pathophysiology is vital for early detection and management [15,16,17,18,19,20]. FDG PET-CT effectively assesses the primary tumor’s response to nCRT and the detection of interval metastases. In cases of true interval metastases, none were associated with progressive or enlarging primary disease [21]. Newly emerged FDG lesions during restaging could result from chemotherapy, radiation, or their combination without pre-therapy signs of metastasis [9,10,22]. The risks of RILD increase with concurrent hepatotoxic chemotherapy, and liver radiation tolerance is reduced in patients with impaired liver function, heightening their risk of RILD [16]. This structured approach aims to navigate the complexities of diagnosing and managing simulated liver metastasis in esophageal cancer patients following nCRT.

## 2. Mechanism and Application of FDG PET-CT Scan

### 2.1. Imaging Principles and Clinical Application

FDG, a non-physiological analog of glucose, differs only slightly from the chemical structure of glucose. It follows the same cellular transport and metabolic pathways [23]. Upon injection, FDG is absorbed by cell membrane glucose receptors—primarily, the glucose transporter-1 molecule (GLUT-1)—which transport it into cells where it is phosphorylated into FDG-6-phosphate by hexokinase. This process traps FDG inside cancer cells, as it fails to undergo further metabolism. This property allows for the visualization of metabolic activity at tumor sites. However, it is well-known that active benign pathological conditions, such as inflammation and infection, can also exhibit increased FDG accumulation. This is due to the enhanced glycolytic metabolism in inflammatory cellular infiltrates, including activated macrophages, monocytes, and polymorphonuclear cells, which play crucial roles in the recruitment, activation, and healing phases of tissue inflammation [24].

The integration of PET-CT using the glucose analog 18F-FDG has become a cornerstone in the imaging of oncological patients. The realization that combining metabolic and morphological information from FDG PET-CT significantly affects tumor staging and restaging, the detection of recurrent disease, and the optimization of therapy across a broad spectrum of solid-organ malignancies has led to its increased adoption in oncology [25,26,27,28]. Specifically, in preoperative evaluation for esophageal cancer, whole-body FDG PET-CT scans are instrumental in assessing the response to radiation treatment and excluding metastases. Furthermore, identifying radiation-induced liver injury (RILI), marked by elevated FDG uptake in areas near the irradiated field, is essential for precise staging and treatment planning [12,13].

### 2.2. False Positive and False Negative PET/CT: Causes and Probabilities

#### 2.2.1. Caveats in Interpreting PET-CT in Individuals with Esophageal Cancer

The interpretation of FDG PET-CT scans in the diagnosis of esophageal cancer necessitates meticulous differentiation between authentic malignant lesions and false-positive signals that may mimic malignancy (Table 1).

#### 2.2.2. Common Non-Malignant Pathological Conditions Showing Increased Uptake of FDG before Therapy

It is estimated that benign, non-physiological lesions with increased FDG uptake are identified in more than 25% of FDG PET-CT studies conducted on oncological patients [29,30,31]. Detecting malignant infiltration of lymph nodes is crucial for the accurate staging of most cancers. FDG PET-CT significantly contributes to this effort by identifying tumor involvement in lymph nodes that are non-pathologically enlarged. These include inflammatory conditions such as sarcoidosis or sarcoid-like reaction to malignancy, collagen vascular diseases, and anthracosis and infective causes such as tuberculosis (TB), infectious mononucleosis, acquired immunodeficiency syndrome (AIDS), and hepatitis C [32,33].

## 3. Current Treatment Protocol of Esophageal Cancer Involving True Liver Metastasis and False Liver Metastasis

### 3.1. Current Standard Procedure of Treatment for Esophageal Cancer (Figure 1)

The typical duration between completing nCRT and undergoing esophagectomy may vary, depending on the specific treatment protocol and the patient’s response to nCRT. Generally, a waiting period of about 4 to 8 weeks is advised to allow patients to recover from the effects of chemoradiation and for any potential tumor downsizing or downstaging to occur. This interval also facilitates an assessment of the patient’s suitability for surgery and allows time for necessary preoperative planning.

nCRT for locally advanced esophageal cancer is a well-established practice prior to surgical resection, as evidenced by the Chemoradiation for Esophageal Cancer Followed by Surgery Study (CROSS) trial [34]. Restaging imaging is recommended to confirm that there has been no interval progression of disease or development of new metastases, which would render the patient unresectable [35]. In esophageal carcinoma, neoadjuvant radiotherapy can be delivered using three-dimensional (3D) conformal therapy or intensity-modulated therapy (IMRT). IMRT has been shown to increase the radiation dose to the primary tumor while minimizing damage to surrounding tissue compared with traditional anterior–posterior opposing field radiotherapy [13]. At restaging with FDG PET-CT after chemoradiotherapy, 8% of patients are found to have interval metastases [9,10]. However, inflammatory reactions can cause false positive results on FDG PET-CT scans [36,37]. RILI is observed in 3–8% of patients reassessed with an FDG PET-CT scan following nCRT. Although this condition is relatively uncommon, being aware of its occurrence is crucial to avoid mistakenly diagnosing metastatic disease (Table 2) [14].

**Figure 1 cancers-16-00948-f001:**
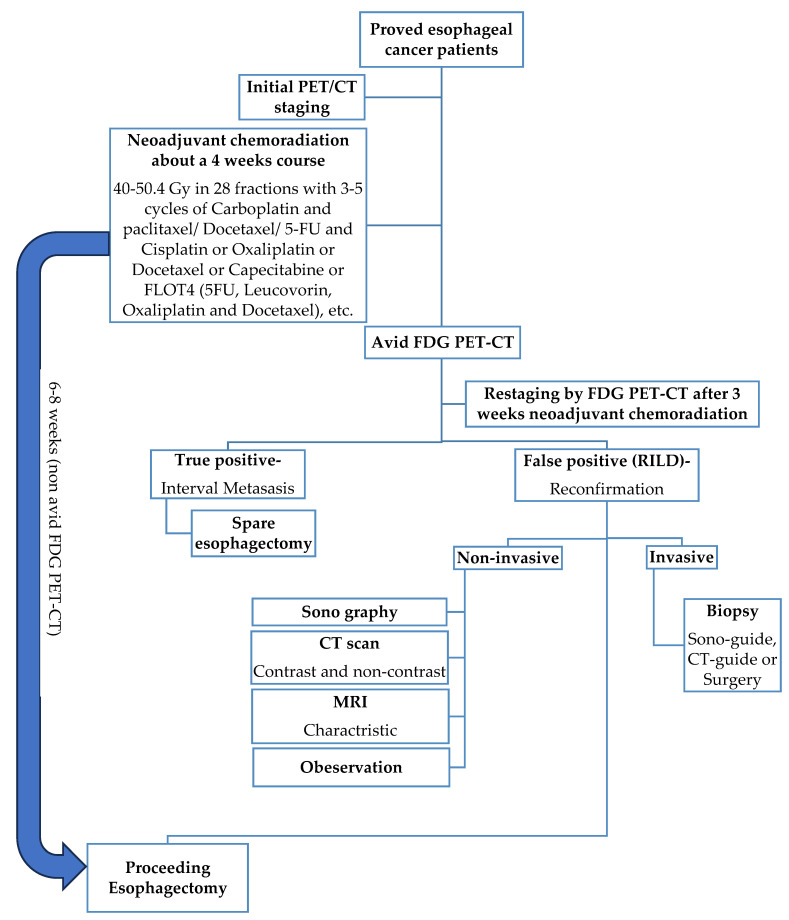
Schematic figure of the current gold standard procedure for esophageal cancer treatment with nCRT for about 4 weeks followed by non-avid FDG PET-CT proceeding esophagectomy.

### 3.2. Reports of the Reference of True and False Metastasis for Restaging after nCRT

A retrospective analysis of 112 patients with distal esophageal cancer, who received nCRT followed by restaging with FDG PET-CT, identified new liver foci in 10 out of 112 patients (9%). Nine of these cases were determined to be RILI, based on further imaging (n = 6) or biopsy (n = 2), and one patient was diagnosed with interval metastatic disease by biopsy. Notably, RILI occurred exclusively in the caudate and left hepatic lobes [22]. Another study involving 26 patients assessed FDG uptake in the liver before and after nCRT for esophageal cancer. New focal FDG uptake in the left liver lobe after chemoradiation was seen in two patients (8%), with no increase in FDG uptake observed in the right part of the liver. A biopsy confirmed radiation injury in one of these patients. CT scans showed atrophy and decreased attenuation in the irradiated left liver lobe in 58% of patients, with no signs of liver metastases [13].

#### Case Reports Series (Table 3) and Cohort Study

The findings of this study reveal that more than 10% of esophageal cancer patients who undergo nCRT have detectable (interval) metastases. With a sensitivity of 73.9% and a specificity of 91.3%, FDG PET-CT proves to be an accurate tool for identifying these cases. Currently, a definitive restaging protocol is still absent, although previous reports have indicated incidence rates of interval metastases between 8 and 17% [8,38,39,40,41]. To date, two reports have specifically addressed the use of FDG PET-CT in detecting interval metastases. In a study by Blom et al., four cases of interval metastases were identified among 50 patients treated neoadjuvantly (8%) [8]. Restaging FDG PET-CT was conducted 6 weeks after completing neoadjuvant therapy, which included 5-FU, cisplatinum, and 50.4 Gy of radiotherapy. This cohort reported a false-positive rate of 2%, and metastatic disease was observed intraoperatively in 1 out of 46 patients (2.2%) [8]. Another study reviewed the records of 85 patients treated either with induction chemotherapy followed by concurrent chemoradiotherapy or with concurrent chemoradiotherapy alone [38]. In this group, post-nCRT FDG PET-CT identified metastatic disease in only 3.9% of patients. The positive predictive value (PPV) of the post-nCRT FDG PET-CT for interval metastases was notably low at 15.6% (10/64), and the findings related to the primary site on post-nCRT FDG PET-CT did not appear to correlate with the development of metastatic disease [21].

The reported study’s findings indicate that 18F-FDG PET-CT restaging after nCRT detects interval metastases in 8% of esophageal cancer patients, with a patient-based sensitivity and specificity of 75% and 94%, respectively. The incidence of interval metastasis in this study aligns with the results from other reports [8,9,10]. However, little is known about the risk factors for developing interval metastases, and the small sample size in the mentioned studies limits the ability to identify predictors for interval metastasis after neoadjuvant therapy [8,9,10,42]. The false positive rate of 6% during 18F-FDG PET-CT restaging was significant, with the lungs and liver being the most commonly affected sites. This supports the literature reports of false positive rates ranging between 0% and 10% [8,43], with the liver and lung as the most frequently involved organs [21,22]. Previous studies evaluating new FDG-avid hepatic lesions within the presumed radiation field in patients with esophageal cancer have shown that these lesions are generally indicative of RILD rather than metastatic disease [13,14,22]. ^18^F-FDG PET-CT restaging accurately detects true distant interval metastases in 8.3% of patients after CRT for esophageal cancer [11].

**Table 3 cancers-16-00948-t003:** Review of case reports.

Author (Year)	Age	TNM- Pathology	Chemotherapy	Radiotherapy Dose- Modality	Delay CRT to FDG PET	FDG PET	CT	MR	Biopsy	Liver Tests	Follow-Up
Iyer et al. (2007) [13]	63	NA-adeno	NA	50.4 Gy-3D conformal	6w	Nodular	Well-defined, low attenuation	-	Perop	AP ↑	NA
Iyer et al. (2007) [13]	NA	NA-NA	NA	50.4 Gy-3D conformal	6w	Nodular	Well-defined, low attenuation	-	NA	AP ↑	NA
Nakahara et al. (2008) [5]	50	uT3N M2 1(bone)-NA	Docetaxel weekly (20 mg/m^2^)	46 Gy + boost 14 Gy-AP-RT	4w	Wedge-shaped	Well-defined, low attenuation + band-like lesion (≈zone of <40 Gy)	-	NA	AP ↑	4 months
DeLappe et al. (2009) [44]	61	uT3N M1 0-NA	4 cycli (apirubicine þ oxaliplatin + capcetabine) + 3 cycli (docetaxel þ irinotecan) + concurrent 5-FU	50.4 Gy-IMRT	5w	Ill-defined nodular	Patchy defined, mixed attenuation, heterogeneous enhancement of left liver	-	CT-guided + perop	NA	NA
Wong et al. (2012) [45]	58	NA-NA	NA	50.4 Gy-AP-RT	6w	Nodular with linear distribution	Patchy-defined, low attenuation in segment 2 and 3	-	NA	Normal	NA
Rabe et al. (2016) [12]	53	uT3N M1 0-squamous	5 cycli (carboplatin + paclitaxel)	50.4 Gy-3D conformal	2w	Nodular	Well-defined, low attenuation	Hyperintens T2-weighted	Perop	AP ↑	12 months
Demey et al. (2017) [46]	42	uT2N M1 0-adeno	Concurrent Oxaliplatin + 5-FU	45 Gy-3D conformal	4w	Nodular	Patchy-defined, low attenuation in segment 2	Hyperintens T2-weighted	Perop	Normal	18 months

NA: data not available; adeno: adenocarcinoma; Gy: Gray; w: weeks; AP: elevated alkaline phosphatase levels; AP-RT: conventional anterior–posterior radiotherapy; IMRT: intensity-modulated radiation therapy; 5-FU: 5-fluorouracil.

## 4. RILD Inducing False FDG PET-CT Interpretations

Clinical radiation injury in the left liver lobe, due to its anatomical position, may affect 6–66% of patients. This variance depends on the volume of hepatic tissue irradiated and the dosage applied [13]. Grant et al. observed that all new 18F-FDG-PET-CT lesions in the right lobe were metastatic, in contrast to the lesions in the left or caudate lobe, which were all radiation-induced injuries. Thus, radiation fields should be meticulously compared with the nodular lesion’s location, and only lesions outside these fields should be considered highly suspicious of metastases [22]. Preoperative 18F-FDG-PET-CT proved useful in the reevaluation of patients after nCRT to determine the treatment response and rule out occult metastasis [46]. Eithne M. DeLappe reported a case of a 61-year-old esophageal cancer patient with increased FDG uptake in the left liver lobe post-50.4 Gy radiation, where no metastasis was found in the biopsy [44]. In Oregon, USA, out of 112 distal esophageal cancer patients undergoing nCRT, 10 showed increased FDG uptake during restaging, with 1 later diagnosed with metastasis, while the rest had RILI [22]. At the Anderson Cancer Center, 26 patients received similar treatment; 2 exhibited increased FDG uptake in the left liver lobe, with no uptake in the right [13]. In a study by Francine et al. involving 205 patients, 6 exhibited increased FDG uptake in the caudate or left lobe during nCRT, but none had liver metastasis [14]. The liver’s sensitivity to radiation restricts the use of radiation therapy for upper abdominal tumors [12,13].

### 4.1. Implications of Increased FDG Uptake

The mechanism behind FDG accumulation in RILI remains uncertain, but it likely involves an inflammatory component. It is well-documented that radiation-induced inflammation, such as postradiotherapy esophagitis [36], shows high FDG uptake due to leukocyte glucose metabolism [5,47]. The exact timing and duration of this phenomenon are unclear. Acute chemoradiotherapy side effects peak during and shortly after treatment and then gradually subside, often taking 6 weeks or more. Serial FDG PET-CT scans for patients suspected of RILI could offer insights into its natural progression [14]. New FDG avidity foci in the liver developed during neoadjuvant therapy in 9% of patients, with 8% identified as having RILD based on further imaging and/or biopsy [22]. Both metastases and RILD can present as increased FDG avidity, usually attributed to RILD during nCRT [22]. nCRT can cause liver damage resembling metastasis on FDG PET-CT, often affecting the caudate and left hepatic lobes [48]. RILD may manifest on PET-CT as focal increased FDG uptake, sometimes mistaken for metastatic disease [48].

### 4.2. The Formation and Classification of RILD

RILD occurs in two forms: classic and non-classic [17]. Classic RILD symptoms appear 1–3 months post-liver radiation therapy (RT), including fatigue, abdominal pain, increased girth, hepatomegaly, and anicteric ascites [18]. Non-classic RILD, often occurring in patients with chronic liver conditions like cirrhosis or viral hepatitis, presents more severe hepatic dysfunctions, such as jaundice and elevated serum transaminases [17]. Radiotherapy targeting the distal esophagus and locoregional nodes can expose adjacent nonmalignant hepatic parenchyma to significant doses, potentially causing acute or chronic radiation hepatitis (RH) [5,13], which may be identified on CT and FDG PET-CT as liver atrophy, attenuation changes, and metabolic abnormalities in the irradiated liver parenchyma [13]. The development mechanisms of RILD remain largely unknown [17], but they likely begin with damage to the central vein and sinusoid endothelial cells, leading to sinusoidal congestion and advanced stages of veno-occlusive disease (VOD) [15,19].

### 4.3. Occurrence and Duration of RILD

Acute RH can occur from a dose of 30 Gy onward, typically manifesting 4–8 weeks after RT completion. Histopathology characterizes it by sinusoidal congestion and fibrosis occluding central veins. Chronic RH emerges over 100 days post-irradiation, marked by portal fibrosis and lobular architecture distortion, without typical centrilobular congestion [5,49]. RILD presents symptoms like anicteric hepatomegaly, ascites, and elevated liver enzymes, usually between 2 weeks to 3 months post-radiotherapy. Most patients recover within 3–5 months, though some may progress to liver fibrosis and failure [14]. Initially described by Reed and Cox, RILD pathophysiology involves retrograde congestion as a key factor [15], with occurrence times ranging from 2 weeks to 7 months post-radiation [16]. About 8% of patients exhibit RILI during restaging [17].

### 4.4. Incidence of RILD

A systematic search reported a 3% incidence of RILI, with retrospective reviews finding higher rates (8%) upon reevaluation of scans for focal uptake in the left liver lobe [13,22]. The discrepancy may be due to undetected cases of RILI in this study. RILD risk escalates with the mean liver dose and irradiated volume. RILD is unlikely with a mean liver dose below 31 Gy [50,51]. In distal esophageal cancer, liver radiation doses typically stay under 30 Gy, but parts of the liver within the radiation target volume may receive up to 40–50 Gy, leading to localized RILI without clinical symptoms [5,52,53]. In a 205-patient study undergoing nCRT, 6 cases showed localized increased FDG uptake in the liver post-nCRT, unrelated to liver metastases. The institute’s RILD incidence was 3%, with the literature citing about 8% at the restaging time [14].

#### 4.4.1. Risk Factors for Radiation-Induced Liver Disease (RILD) by Tumor Cell Type

Distal esophageal cancers, including SCC and adenocarcinoma, present distinct risk factors (Table 4) [54]. Alcohol consumption, a major risk factor for SCC, suggests patients with SCC might face a higher incidence of liver dysfunction and, consequently, RILD, compared with those with adenocarcinoma. This area, however, remains under-researched, indicating a need for further studies to elucidate these correlations better.

#### 4.4.2. Influence of Gender, Age, and Race on RILD Susceptibility

Although esophageal cancer predominantly affects males, the specific sensitivities of different genders, ages, and ethnicities to RILD are not well-defined in the current literature. The absence of detailed statistical analysis on these demographic aspects underscores a crucial area for future research attention (Table 5).

### 4.5. Molecule Biology of RILD

Recent advancements in the understanding of RILD pathogenesis have yet to elucidate its molecular pathways fully. The pathogenesis of RILD involves vascular changes, increased collagen synthesis, and the activation of growth factors and cytokines such as TNF-α, TGF-β, and Hedgehog, which are significant in liver repair [17,55]. Radiation exposure results in DNA damage, oxidative stress, and the production of reactive oxygen species, leading to hepatocellular apoptosis and inflammatory responses. Notably, Kupffer cells (KCs) increase the susceptibility of hepatocytes to radiation-induced apoptosis through TNF-α secretion. Hepatic stellate cells (HSCs) also play a pivotal role by transdifferentiating into myofibroblastic HSCs, the primary collagen-producing cells in the liver, upon radiation exposure. This transdifferentiation is key in the development of RILD given the high radiosensitivity of these cells. Furthermore, sinusoidal endothelial cell (SEC) apoptosis is recognized as a primary event in RILD [55].

### 4.6. Effects and Implications of RILD

#### 4.6.1. Radiotherapy Dosage

RILD radiation dose reports in the literature vary, mentioning pure doses up to 70 cGy, combined with 30 cGy chemotherapy, or adjusted doses for pre-existing liver disease. The risk of RILD strongly correlates with the mean liver dose and the volume of irradiated liver. In distal esophageal cancer, parts of the liver may receive doses up to 40–50 Gy, leading to localized RILD without clinical symptoms [14]. Various radiation doses and chemotherapy regimens were used in different cases, including 41.4 Gy, 50 Gy, and 50.4 Gy in conjunction with chemotherapy agents like carboplatin and cisplatin [14].

#### 4.6.2. Synergistic Effects with Chemotherapy

The combined effects of chemotherapy and radiotherapy, particularly when using 3D conformal or intensity-modulated therapy (IMRT) techniques, can induce complex liver responses. Understanding these synergetic effects and the liver’s radiation tolerance is crucial for predicting and managing potential RILD [9,10,16,22,56,57]. No specific details were provided in the documents.

#### 4.6.3. Pre-Existing Liver Diseases and Increased RILD Vulnerability

Information regarding Hepatitis B or C virus infection, liver cirrhosis, hepatomegaly, and liver function impairment were not specified in the documents. Hepatitis B virus carriers have a higher susceptibility to RILD [20]. Hepatitis C is presumed to pose a similar risk level to Hepatitis B, necessitating further research to comprehend the implications for patients with liver function risk factors associated with these hepatitis types.

### 4.7. Challenges in Diagnosing RILD Using Imaging

Differentiating RILD from metastatic disease using imaging techniques such as FDG PET-CT and MRI presents significant challenges. Although imaging features are evolving, they currently offer vital insights into the nature and extent of radiation-induced damage [44,52,58,59,60,61,62]. No specific details were found regarding the sensitivity of Sono, CT, MRI, FDG PET-CT liver, or liver biopsy techniques in the documents.

#### 4.7.1. Sonography of Liver and Its Sensitivity

Ultrasonography remains a mainstay for anatomical imaging of the liver. The advent of new techniques, such as elastography and quantitative ultrasound parameters, has broadened the scope for assessing liver tissue properties beyond mere echogenicity. This involves measuring acoustic parameters to gauge tissue microstructure, which shows promise in monitoring the severity of hepatic steatosis in chronic liver diseases. Such advancements in ultrasound technology significantly enhance the diagnosis of liver disease, particularly in identifying RILD [63]. Sonography typically reveals a hypoechoic appearance over the caudate lobe (Figure 2A).

#### 4.7.2. CT Scan and Its Sensitivity

The evolution of noninvasive imaging techniques continues to refine RILD characterization [58]. Post-radiotherapy CT scans reveal reversible, distinct areas of reduced enhancement in irradiated liver regions, possibly indicating an increase in water or fat content [52,59,60]. Radiation-induced VOD may cause enhanced imaging due to increased augmented arterial flow or delayed contrast clearance [61]. RILD can manifest as hypo- or hyper-attenuated non-anatomic areas [62], with CT imaging typically displaying sharp, straight margins aligned with radiation portals [13]. In contrast, metastatic lesions tend to appear more mass-like and rounded on CT scans [13]. Acute RH is characterized by areas of low attenuation with sharp linear borders on non-contrast CT, observable in patients receiving more than 30–45 Gy [5,12,13,45]. Enhanced CT imaging may show increased contrast in irradiated liver areas due to augmented arterial flow or delayed contrast clearance from radiation-induced VOD [12]. Contrast CT scans depict decreased enhancement in S1 of the liver (Figure 2B), indicating edema in the irradiated area. Modern radiotherapy techniques, utilizing multiple beams from varied angles, present a less pronounced dose gradient. This results in a small volume of normal tissue near the target area receiving a relatively high dose, while a larger volume of surrounding tissue receives lower doses. Consequently, liver injury is typically confined to the high-dose irradiated area, leading to localized edema and reduced attenuation on CT scans [64].

#### 4.7.3. MRI and Its Sensitivity

MRI imaging post-liver radiation showcases decreased signal intensity on T1-weighted images, increased T2 intensity, and enhanced proton spectroscopic imaging signals in irradiated lobes, indicating elevated water content [60]. The high resolution and soft tissue contrast of MRI make it ideal for differentiating organs [65]. Clinical studies have used MRI to monitor radiation damage in the liver [66], myocardium [67,68], and bone marrow [69,70]. MRI performed five weeks post-RT revealed a hypointense signal on T1-weighted images and a markedly hyperintense signal on T2-weighted images, alongside facilitated diffusion on diffusion-weighted MR imaging (DWI) images. These findings showed heterogeneous alterations in the entire left liver lobe, attributed to mild RH in this region and central acute RH in segment 2 [46]. The manifestation of MRI T1-weighted images of the liver displayed low signal intensity over the caudate lobe (Figure 2C), whereas whole T2-weighted images showed strong signal intensity over the same area (Figure 2D). Generally, liver areas subjected to high radiation doses exhibit low signal intensity on T1-weighted images and high signal intensity on T2-weighted images due to edema [37,64].

#### 4.7.4. SUVmax (Standardized Uptake Value) Value in FDG PET-CT Serve as Indicators of RILD

The reported studies do not extensively compare FDG PET-CT SUVmax value between radiation injury and metastatic lesions. However, RILD typically shows SUVmax ranging from 4 to 9/h, while metastatic lesions often have values exceeding 10/h, suggesting the presence of metastasis. New hepatic FDG foci observed during neoadjuvant chemoradiation for esophageal cancer usually signify RILD, attributed to the increased FDG uptake by active leukocytes involved in inflammatory responses [44]. This pattern implies a lower likelihood of metastasis [22]. The location of these foci within the radiation field, typically the left and caudate lobes, is a crucial factor [22]. Before undergoing nCRT, the esophageal tumor demonstrated high FDG uptake (measuring 9.7 × 5.6 cm with an SUVmax of 29.3/h), as indicated by red circles. Notably, there were no active lesions in liver segment I prior to nCRT, as indicated by red arrows (Figure 3). Imaging showed a reduction in esophageal tumor after six weeks of nCRT (measuring 2.1 × 1.6 cm with an SUVmax of 7.7/h), as highlighted in yellow circles (Figure 4). Six weeks post-nCRT, a new FDG-avid lesion appeared in liver segment I (measuring 3.5 × 1.5 cm with an SUVmax of 4.2/h), as indicated by yellow arrows (Figure 5).

#### 4.7.5. Biopsy Options for Diagnosis of Liver lesions—Procedures Guided by Sonography, CT Scan, Mini-laparoscopy, Open Biopsy, or Clinical Observation

##### Pathological Characteristics of RILD

RILD manifests as a VOD primarily affecting the central veins [71]. The process begins with radiation-induced damage to endothelial cells, leading to platelet activation and fibrin deposition. This cascade results in vessel congestion, activation of hepatic stellate cells, and obstruction of blood flow. Consequently, these events trigger the loss of hepatocytes, fibrosis, and potentially necrosis [71]. 

##### Gross and Microscopy Appearance of RILD

The gross and microscopic examination of RILD reveals significant pathological alterations. The affected liver tissue, particularly in the caudate lobe, appears dark red and infiltrated with blood, indicating acute radiation damage (Figure 6A,B). Microscopically, this damage is characterized by congestion, thinned hepatic cords, and spaces filled with erythrocytes (Figure 6C,D). These findings disrupt the normal liver architecture, signaling extensive damage to the liver parenchyma due to vascular and cellular reactions to radiation. Such damage impairs liver function by disturbing blood flow and causing cell death. The specific reference to the caudate lobe suggests a localized impact of radiation, providing insights into the radiation’s distribution and intensity.

## 5. Overview of Literature Review

A comprehensive summary of various studies related to the specific study and findings of esophageal cancer treatment is provided in Table 5.

**Table 5 cancers-16-00948-t005:** Overview of Literatures.

	Author (Year)	Gender	Age (Range)	Race	Chemoradiotherapy			Liver Function *	Stage	Esophageal Cancer		
					Neoadjuvant	Dose	Medicine			SCC	Adeno	Other
1	Rabe et al. (2016) [12]	F	53	NA	**Yes**	50.4 Gy	Cbp and Ptx	Yes	T3N1M0-->T2-weighted	1	0	0
2	Iyer et al. (2007) [13]	24M/2F	54 (41–78)	NA	**Yes**	50.4 Gy	NA	Yes	NA	2	24	0
3	Daly et al. (2007) [4]	74.2%M/25.8%F, n = 5044	67.3	76.8% Non-Hispanic Caucasian, 19.2% African American, 4.0% Hispanic	NA	NA	NA	NA	Clinical stage—0 (2.2%), I (14.1%), II (23.0%), III (22.1%), IV (38.7%)	51.6%	41.9%	0
4	Nakahara et al. (2008) [5]	M	50	NA	**Yes**	46 Gy with an additional boost irradiation of 14 Gy.	Dot	Yes	Diagnosed with esophageal cancer with lymph node and bone metastases	NA	NA	0
5	Voncken et al. (2018) [14]	M	50	NA	**Yes**	50.4 Gy	Cbp and Ptx	NA	T3N1M0	1	0	0
		M	62	Not specified	**Yes**	41.4 Gy	Cbp and Ptx	No	T3N0M0	0	62	0
		M	41	NA	**Yes**	41.4 Gy	Cbp and Ptx	No	T3N1M0	0	41	0
		M	59	NA	**Yes**	50 Gy	Cis and 5-FU	No	T3N1M0	0	1	0
		M	49	NA	**Yes**	41.4 Gy	Cbp and Ptx	No	T3N1M0	0	1	0
6	Stiekema et al. (2014) [10]	60M/16F	63 (46–80)	NA	**Yes**	50 Gy or 50/50.4 Gy	5-FU and Cis or Cbp and Ptx	NA	NA	14	60	2
		24M/2F	63 (46–80)	NA	**Yes**	50 Gy (n= 21) or 41.4 Gy (n = 50) or 50.4 Gy (n = 5)	5-FU and Cis (n = 21) or Cbp and Ptx (n= 55)	NA	NA	9	39	0
7	Grant et al. (2014) [22]	93M/19F	57 (28–81)	NA	**Yes**	41.4–50.4 Gy	NA	NA	NA	21	97	4
8	Wieder et al. (2004) [56]	27M/11F	60 (46–73)	NA	**Yes**	40 Gy	5-FU	NA	NA	38	0	0
9	DeLappe et al. (2009) [44]	M	61	NA	**Yes**	50.4 Gy	NA	NA	T3N1M0	0	1	0
10	Shai et al. (2020) [48]	M	66	Asian	**Yes**	50 Gy	NA	No	T3N1M0	1	0	0
11	Demey et al. (2016) [46]	M	42	NA	**Yes**	45 Gy	Oxa, levofolinic acid, and 5-FU	No	uT2N1M0	0	1	0
12	Anderegg et al. (2015) [72]	76.3%M, n = 156	65 (34–83)	NA	**Yes**	41.4 Gy	Cbp and Ptx (n = 139) or Cbp, Ptx, and Vectibix (n = 17)	NA	NA	29	126	1
13	Voncken et al. (2018) [14]	M	50	NA	**Yes**	50.4 Gy	Cbp and Ptx	NA	T3N1M0	1	0	0
		M	62	NA	**Yes**	41.4 Gy	Cbp and Ptx	No	T3N0M0	0	1	0
		M	41	NA	**Yes**	41.4 Gy	Cbp and Ptx	No	T3N1M0	0	1	0
		M	59	NA	**Yes**	50 Gy	Cis and 5-FU	No	T3N1M0	0	1	0
		M	49	NA	**Yes**	41.4 Gy	Cbp and Ptx	No	T3N1M0	0	1	0
		M	75	NA	**Yes**	50 Gy	Cbp and etoposide	No	T2N1M0	0	0	1
14	Goense et al. (2018) [11]	675M/108F	<65, n = 425; ≥65, n = 358	NA	**Yes**	45 Gy or 50.4 Gy	Oxa and 5-FU or Doc and 5-FU or Xeloda and 5-FU or other	NA	NA	111	672	0
15	Gabriel et al. (2017) [21]	234M/24F	61.5	NA	**Yes**	50.4 Gy	Cis and Iri/Cbp and Ptx or Oxa and Xeloda or 5-FU and Cis	NA	NA	39	219	0
16	Li et al. (2020) [73]	76M/48F	56 (25–82)	NA	NA	NA	NA	NA	NA	20	69	35
17	Blom et al. (2011) [8]	40M/10F	61 (56–67)	NA	**Yes**	50.4 Gy	Cis and 5-FU	NA	Stages II to IVa	9	40	1
18	Cerfolio et al. (2005) [40]	41M/7F	68 (48–76)	NA	**Yes**	<50 Gy (n = 22), >50 Gy (n = 26)	NA	NA	Stages I to Ivb	5	43	0

M: male; F: female; NA: data not available; SCC: squamous cell carcinoma; Adeno: adenocarcinoma; Gy: Gray; Cbp: Carboplatin; Ptx: paclitaxel; Dot: Docetaxel; 5-FU: 5-fluorouracil; Cis: cisplatin; Iri: irinotecan; Oxa: oxaliplatin; Xeloda: Capecitabine; Vectibix: panitumumab. * Liver function: Yes indicates abnormal; No represents normal.

## 6. Conclusions

New foci of increased FDG avidity are commonly observed in the caudate and left hepatic lobes of the liver during nCRT for distal esophageal cancer. These findings are often indicative of RILD rather than metastatic disease. It is crucial to be aware of the pitfalls associated with high FDG uptake in RILI to prevent misinterpretation and ovestaging. In addition to the location of FDG uptake, the lesion’s shape, and an SUVmax value greater than 10/h, a convincing liver MRI or even a liver biopsy can provide accurate information to distinguish between RILI and liver metastasis. Typically, surgery is scheduled for approximately 6–8 weeks after the completion of CRT. Prior to proceeding with surgery, an FDG PET-CT evaluation is recommended to check for new interval metastases. Patients presenting with these will usually not proceed to surgery.

## Figures and Tables

**Figure 2 cancers-16-00948-f002:**
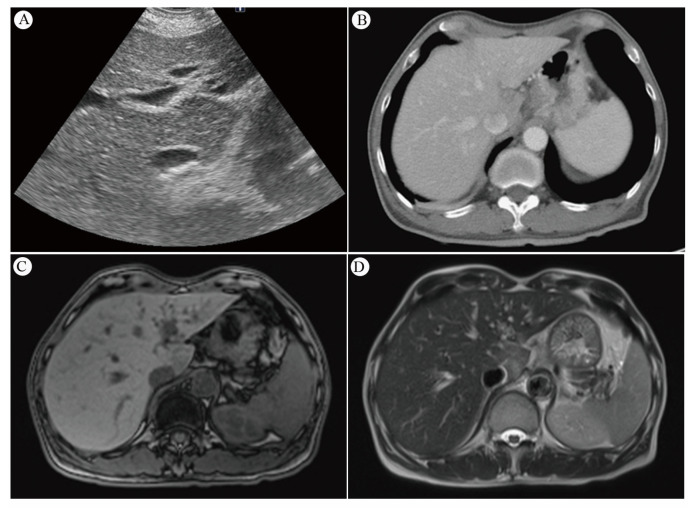
(**A**) Sonography of the liver reveals a hypoechoic appearance over the caudate lobe. (**B**) The contrast CT scan shows decreased enhancement in S1 of the liver. (**C**) *MRI* T1-weighted pictures of the liver reveal a low signal intensity over the caudate lobe. (**D**) *MRI* T2-weighted pictures of the liver indicate a strong signal intensity over the caudate lobe. Reprinted with permission from Sen-Ei Shai et al. (2020) [48].

**Figure 3 cancers-16-00948-f003:**
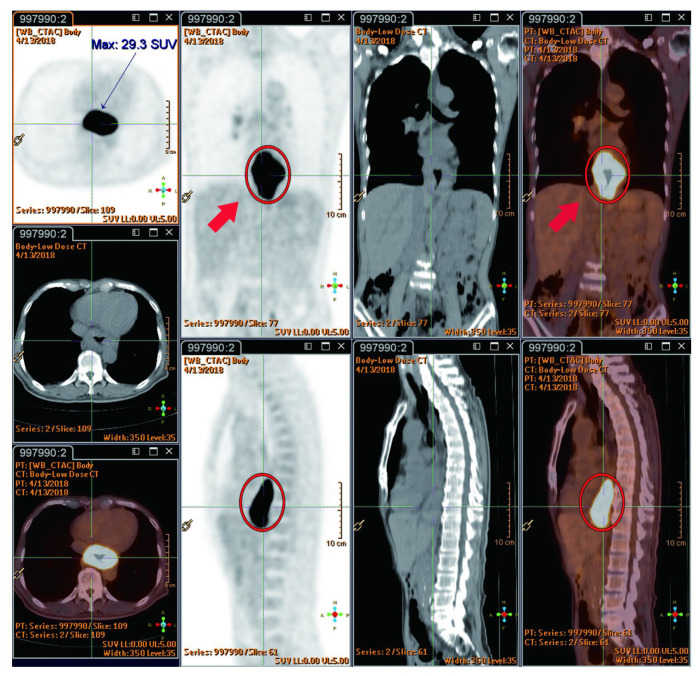
Prior to neoadjuvant chemoradiotherapy (nCRT), the esophageal tumor exhibited high FDG uptake (9.7 × 5.6 cm, SUVmax: 29.3/h) (red circles). There were no active lesions in liver segment I before nCRT (red arrows). Reprinted with permission from Sen-Ei Shai et al. (2020) [48].

**Figure 4 cancers-16-00948-f004:**
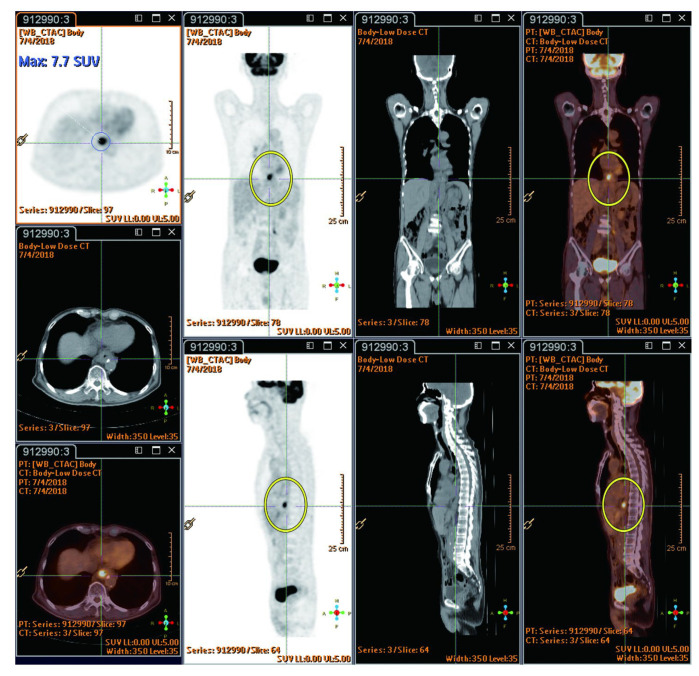
Shows esophageal tumor reduction after six weeks of nCRT (2.1 × 1.6 cm, SUVmax: 7.7/h, highlighted by yellow circles). Reprinted with permission from Sen-Ei Shai et al. (2020) [48].

**Figure 5 cancers-16-00948-f005:**
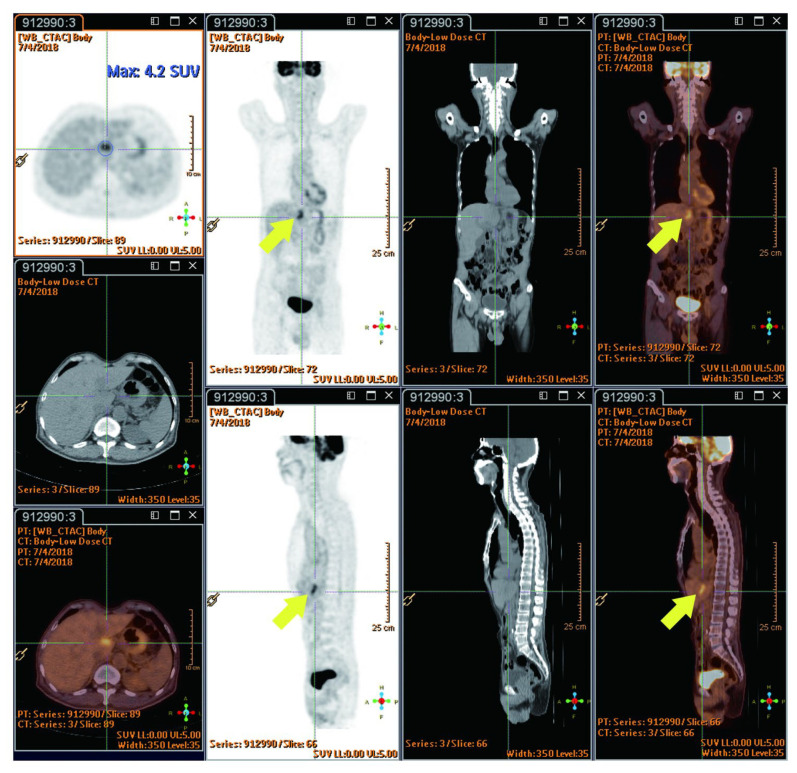
Six weeks following nCRT, a new FDG-avid lesion was found in liver segment I (3.5 × 1.5 cm, SUVmax: 4.2/h, indicated by yellow arrows). Reprinted with permission from Sen-Ei Shai et al. (2020) [48].

**Figure 6 cancers-16-00948-f006:**
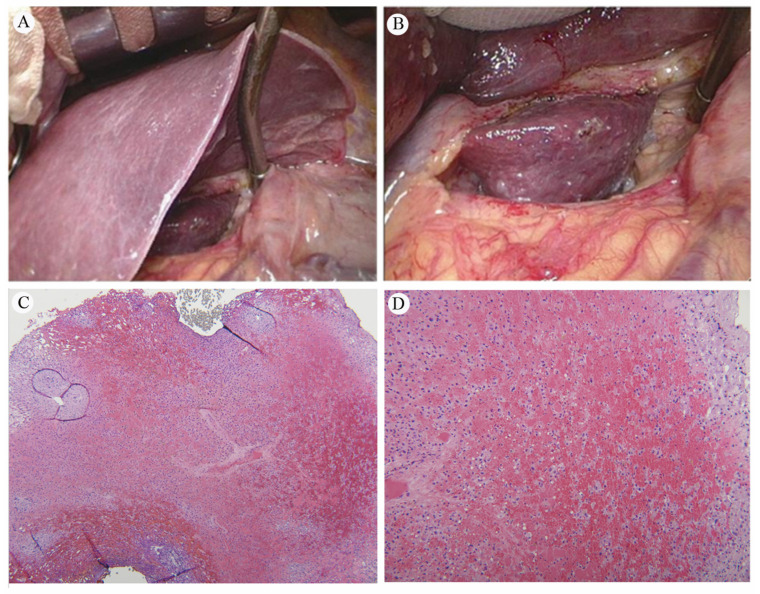
(**A**) The normal liver versus the inflamed caudate lobe. (**B**) An up-close image of the liver, showing dark red, soft tissue, and blood infiltration in the caudate lobe. Pathology of the liver caudate lobe. (**C**) A low-power field reveals no tumor metastasis at a magnification of 20×. (**D**) A high power field indicates congestion with attenuated hepatic cords filled with erythrocytes at a magnification of 40×. Reprinted with permission from Sen-Ei Shai et al. (2020) [48].

**Table 1 cancers-16-00948-t001:** Caveats in the interpretation of PET-CT in patients with esophageal cancer.

Causes of False-Positive Findings	Causes of False-Negative Findings
Infections/inflammatory lesions	Lesion dependent
Radiation-induced liver disease (RILD)	Small tumors (<8–10 mm)
Radiation pneumonitis	Low metabolic activity of the tumor
(Postobstructive) pneumonia/abscess	The presence of a treatment-induced decrease in tumor metabolism
Mycobacterial or fungal infection	Technique limitation
Granulomatous disorders (sarcoidosis, Wegener)	Hyperglycemia
Chronic nonspecific lymphadenitis	Paravenous FDG injection
(Rheumatoid) arthritis	Excessive time between injection and scanning
Occupational exposure (anthracosilicosis)	Low resolution or motion artifacts
Bronchiectasis	
Organizing pneumonia	
Reflux esophagitis	
Iatrogenic causes	
Invasive procedure (puncture, biopsy)	
Talc pleurodesis	
Radiation esophagitis and pneumonitis	
Bone marrow expansion postchemotherapy	
Colony-stimulating factors	
Thymic hyperplasia postchemotherapy	
Benign mass lesions	
Salivary gland adenoma (Whartin)	
Thyroid adenoma	
Adrenal adenoma	
Colorectal dysplastic polyps	
Focal physiological FDG uptake	
Gastrointestinal tract	
Muscle activity	
Brown fat	
Unilateral vocal cord activity	
Arherosclerotic plaques	

**Table 2 cancers-16-00948-t002:** Diagnostic parameters of 18F-FDG PET-CT for the detection of interval metastasis by Goense et al., 2018 [11].

Parameter	18F-FDG PET-CT
Sensitivity (%) [95%CI]	65/87 (74.7%) [64.3–83.4]
Specificity (%) [95%CI]	652/696 (93.7%) [91.6–95.4]
Positive predictive value (%) [95%CI]	65/109 (59.6%) [52.0–66.9]
Negative predictive value (%) [95%CI]	652/674 (96.7%) [95.4–97.7]
Diagnostic accuracy	91.6%

**Table 4 cancers-16-00948-t004:** Risk factors for esophageal cancer * from Enzinger et al., 2003 [54].

Risk Factor	Squamous Cell Carcinoma	Adenocarcinoma
Tobacco use	+++	++
Alcohol use	+++	-
Barrett’s esophagus	-	++++
Weekly reflux symptoms	-	+++
Obesity	-	++
Poverty	++	-
Achalasia	+++	-
Caustic injury to the esophagus	++++	-
Nonepidermolytic palmoplantar keratoderma (tylosis)	++++	-
Plummer–Vinson syndrome	++++	-
History of head and neck cancer	++++	-
Frequent consumption of extremely hot beverages	+	-

* A single plus sign indicates an increase in the risk by a factor of less than two, two plus signs indicate an increase by a factor of two to four, three plus signs indicate an increase by a factor of more than four to eight, and four plus signs indicate an increase by a factor of more than eight.

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
