# Peer review of "False Liver Metastasis by Positron Emission Tomography/Computed Tomography Scan after Chemoradiotherapy for Esophageal Cancer—Potential Overstaged Pitfalls of Treatment"

_cancers, 2024, doi:10.3390/cancers16050948_

Round 1

Reviewer 1 Report

Comments and Suggestions for Authors

Dear authors, 

this is a really interesting work focusing on an important topic of the treatment of cancer patients.

The main issue that I noticed is that it is not clear if this is a narrative review (in this case the type of paper at the top of the first page should be changed) or a systematic review (in this case some important part such as search strategy, paper selection, quality assessment and so on are missing). Please explain this point. 

Author Response

Dear reviewer,

Thanks for your correction. This is a narrative review instead of systemic review, so we spare search strategy, paper selection, quality assessment. Thanks for your reminding.

Reviewer 2 Report

Comments and Suggestions for Authors

Interesting careful review on a hot topic. I have only one single point to be discussed in the revised version.

Minor point: 

Is mini-laparoscy an option to evauluate the left liver lobe perhaps accompanied by liver biopsy in case of uncertaincy?

Comments on the Quality of English Language

minor English improvement is needed.

Author Response

Dear reviewer,

Thanks for your suggestion. We will elaborate the manuscript and mark the revision. Thanks for your reminding.

Minor point: Is mini-laparoscy an option to evauluate the left liver lobe perhaps accompanied by liver biopsy in case of uncertaincy?

ANS: Thanks for your excellent correction. Mini-laparoscopy is good option for tissue biopsy, we will add this in “4.7.5. Biopsy for Diagnosis of Liver- Options by Sono-guiding, CT-guiding, Mini-laparoscopy, Open biopsy or Clinical Observation”. Thanks again.

Reviewer 3 Report

Comments and Suggestions for Authors

This paper is a review regarding challenging pitfalls in detection of new interval liver metastases in post neo-adjuvant chemoradiotherapy for oesophagus cancers with FDG PET-CT. Subject is interesting and important.

Major comments:

Methodology about litterature analysis is not described and finally quite confuse.

3.2.1.: 

Line 168: which is the current study? Have results of previous studies been merged ?

Line 187 and Line 275 (4.4.): same question.

Minor comments:

Simple summary: 

CRT: please define.

2.2.3.:

TB: please define.

Figure 1 :

Maybe adding non avid PET-CT after “nCRT about 4weeks course” and before Proceeding Esophagectomy”?

Adding FDG before “PET-CT” each time expression is used.

Table 4 (the first one… : please correct):

Please rule out reference numbers (that do not bind with this paper’s references numbers).

4.:

Line 223 to 226: please rewrite without intermediate title “Hepatic Radiosensitivity”.

4.1.:

Line 236: same remark.

4.4.1.: 

Title: weird formulation.

Table 4 (the second one…):

Plus-minus is not used : is this an oversight ?

4.6.1.:

cGY : please write cGy.

4.6.3.: 

Title: weird formulation.

4.7.4.: 

SUVmax notation: /h? /1h? Does it mean “one hour after injection”? If so, please rewrite ( /h means “per hour”).

Line 412 : unclear.

Where are supplementary materials ?

Comments on the Quality of English Language

A few strange formulations along the text (please refer to minor comments).

Author Response

Dear reviewer,

Thanks for your comments. We will revise and response point-by-point. Thanks for your reminding.

Major comments:

Methodology about litterature analysis is not described and finally quite confuse.

3.2.1.: 

Line 168: which is the current study? Have results of previous studies been merged?

Line 187 and Line 275 (4.4.): same question.

ANS: Thanks for your correction, we revise them and delete the “current” in the main text. Thanks again.

Minor comments:

Simple summary: 

CRT: please define.

ANS: Thanks for your reminding, we revise and define CRT (chemoradiotherapy) in abbreviation.

2.2.3.:

TB: please define.

ANS: Thanks for your reminding, we revise and define TB (tuberculosis) in abbreviation.

Figure 1 :

Maybe adding non avid PET-CT after “nCRT about 4weeks course” and before Proceeding Esophagectomy”?

Adding FDG before “PET-CT” each time expression is used.

ANS: Thanks for your excellent suggestion, we add FDG before “PET-CT” and revise the Figure1 and legend with “…non avid FDG PET-CT proceeding esophagectomy”.

Table 4 (the first one… : please correct):

Please rule out reference numbers (that do not bind with this paper’s references numbers).

ANS: Thanks for your kind correction, we delete the reference number 56.

4.:

Line 223 to 226: please rewrite without intermediate title “Hepatic Radiosensitivity”.

4.1.:

Line 236: same remark.

ANS: Thanks for your brilliant suggestion, we revise as” The liver's sensitivity to radiation restricts the use of radiation therapy for upper abdominal tumors”.

4.4.1.: 

Title: weird formulation.

ANS: Thanks for your great suggestion, we revise to “Risk Factors for Radiation-Induced Liver Disease (RILD) by Tumor Cell Type”.

Table 4 (the second one…):

Plus-minus is not used : is this an oversight ?

ANS: Thanks for your correction, we delete “Plus-minus…” in Table 4.

4.6.1.:

cGY : please write cGy.

ANS: Thanks for your correction, we revise “cGY “ to “cGy” in the main text.

4.6.3.: 

Title: weird formulation.

ANS: Thanks for your great suggestion, we revise to “Pre-existing Liver Diseases and Increased RILD Vulnerability”.

4.7.4.: 

SUVmax notation: /h? /1h? Does it mean “one hour after injection”? If so, please rewrite ( /h means “per hour”).

ANS: Thanks for your kind correction, we revise to /h in the main text.

Line 412 : unclear.

ANS: Thanks for your suggestion, we revise the description as “The manifestation of MRI T1-weighted images of the liver display low signal intensity over the caudate lobe (Figure 2C)”in MRI section.

Where are supplementary materials ?

ANS: Thanks for your correction, we delete this part of declaration.

Round 2

Reviewer 1 Report

Comments and Suggestions for Authors

All the issues that were present in the previous version have been addressed.

Reviewer 3 Report

Comments and Suggestions for Authors

Understanding this is a narrative review, manuscript is now easy-readable.